# Barriers to conducting implementation science research in Asia: An online survey

**Wen Ting Tong** [ID][1]*, **Chor Yau Ooi**[2], **Yee En Lim** [ID][3]

1 Department of Primary Care Medicine, Faculty of Medicine, Universiti Malaya, Kuala Lumpur, Malaysia,
2 Faculty of Medicine and Health Sciences, Universiti Malaysia Sarawak, Kuching, Sarawak, Malaysia,
3 Faculty of Medicine, Universiti Malaya, Kuala Lumpur, Malaysia

* tongwenting@um.edu.my

## Abstract

### Background

Implementation science (IS) has importance in Asia to facilitate effective delivery of evidence-based interventions to bridge disease prevention and management gaps. However, IS research conducted in Asia are slower compared to the Western region.

### Purpose

This study aims to assess barriers to conducting IS research in Asia.

### Methods

This study adopted a cross-sectional online survey design. Researchers who had conducted IS research in Asian settings completed a questionnaire which examined socio-demographic characteristics, IS research practices, and yes or no items on the various barriers to conducting IS research. An open-ended question was provided to gather additional barriers.

### Results

The response rate for this study was 40.1% (113/282). More than half of the participants has PhD level education (68.1%, n = 77), and 46.0% (n = 52) were academic lecturers. The most common barrier to conducting IS research was the lack of funding (86.7%, n = 98) followed by the lack of awareness (59.5%, n = 67), the lack of time (56.6%, n = 64), and the lack of organisational support (48.7%, n = 55). The open-ended responses elicited additional barriers: the lack of commitment and involvement from stakeholders other than researchers, perceived low value of IS research, the lack of scientific rigour as a field, existing knowledge, system and practices challenge, the lack of capacity or training support, and the lack of academic progression indicators.

**Data availability statement:** All data are available from the Open Science Framework database (DOI: 10.17605/OSF.IO/CBSNV).

**Funding:** This study was funded by the Universiti Malaya Research Excellence Grant (UMREG016_2023). The funder of this study had no role in study design, data collection and analysis, decision to publish, or preparation of the manuscript.

**Competing interests:** The authors have declared that no competing interests exist.

## Conclusions

To facilitate the increase of IS research in Asia, more support are needed in terms of availability of funding, capacity building and training in the region, and effective ways to build constructive long-term collaborations with stakeholders.

## Introduction

Every day, research findings are being produced, however, their uptake in real-world practice remains a challenge. It has been reported that it takes an estimated 17 years for research findings to be translated into clinical practice [1–3], although in recent time, this delay has improved from 9 to 13 years in the field of mental health and cardiovascular diseases [4]. Delays in translation has raised concerns in view of waste of expensive resources, and a sacrifice of potential patient benefit [5]. Research cannot contribute to effective and efficient healthcare unless it reaches its intended users and are routinely adopted into practice.

Implementation science (IS) is a field of study that can help to bridge the knowledge-practice gap. It is defined as *"the scientific study of methods to promote the systematic uptake of research findings and other evidence-based practices into routine practice, and hence, to improve the quality and effectiveness of health services"* [6]. In the last decade, the field has advanced rapidly and a wide range of research studies have been conducted including exploring barriers and facilitators to implementation, exploring influences on the provider, patient, and organizational behaviours, testing implementation strategies' effectiveness, developing theories and frameworks to support implementation, and developing and validating measures to assess implementation outcomes [7]. However, most of the IS literature are from Western high-income countries such as the USA, UK, Australia, and Canada [8–10]. Nevertheless, some IS work has been observed in low- and middle-income countries particularly from the Africa region including the adaption of IS frameworks, and the selection and application of implementation strategies [11,12]. The advancement of IS research in the African region can be attributed to the support by Western-country partners or international organisation, which provided funding, and cross-country learning opportunities for IS training and capacity building in the region [13,14], although the lack of funding, collaborators and mentorship prevailed as main barriers to engage in IS research [14].

IS has particular importance in Asian settings as it is the region where many low- and middle-income countries reside. The region experiences higher mortality and poor health outcomes when face with global health burdens due to limited resources as compared to high-income countries [15]. Improving the time and effective delivery of evidence-based interventions can help to bridge disease prevention and management gaps. IS provides approaches and tools to develop strategies that can generate and/or maximize resources to facilitate the implementation of evidence-based interventions in resource-constrained settings [16].

It has been 17 years since IS emerged, with the launch of the first dedicated specialist scientific journal, *Implementation Science* in 2006 [6]. However, although IS research in Asia is still limited, it is gradually gaining traction [17]. Based on narrative reviews and commentaries, the slow adoption of IS research in Asia is said to be attributed to the lack of emphasis by policymakers in setting research agendas or fund implementation research, the lack of applicability of IS theory/model/framework in low and middle resource settings, the monopoly of knowledge sharing by high-income countries, the lack of emphasis of equity and community participation in the implementation research agenda, and the lack of IS expertise and basic understanding of IS [18,19]. While there has been a call for increasing IS research in the Asia [18], there is a lack of initiatives in advancing the field in the region. Hence, understanding barriers to conducting IS research can increase awareness, and help to identify potential strategies to promote the conduct of IS research further. Currently, understanding of barriers to IS research in the Asian settings has not been empirically studied. Therefore, this study aims to assess barriers to conducting IS research in Asia.

## Methodology

### Study design

This study adopted a cross-sectional online survey design. This study design was chosen to provide current insights on the various barriers to conducting IS research in the Asian region. The wide study scope that involves the Asian region necessitates the use of online survey as it is cost effective and provide convenience for participants who may be from different time zones to complete the survey anytime, and anywhere.

### Participants

Researchers who had conducted IS research in Asian settings were invited to participate in this study. These participants were identified from implementation science studies that were published, whereby the corresponding authors of the studies were invited to participate. The study sampling frame was obtained via an online literature search on IS research conducted in Asian settings using the PubMed, EMBASE, CINAHL, and PsycINFO databases. More information on the search process and strategy can be found in S1 File.

The inclusion criteria for the articles were:

- study that use an implementation theory, model, framework in guiding the implementation effort [20], perform a process or implementation evaluation, or described use of any of the 73 implementation strategies (ERIC), or presented findings on implementation outcomes (i.e., acceptability, adoption, appropriateness, feasibility, fidelity, implementation cost, penetration and sustainability) [21]

- conducted in Asian settings

- studies conducted by non-Asian researchers, but the study was conducted in an Asian setting

- original research (rather than a letter, commentary, protocol, editorial, review)

- was authored by a named (rather than an anonymous) author

- published from 2006 onwards

- published in English

The exclusion criteria were:

- studies conducted in multi-countries (including both Asian and non-Asian countries).

- studies conducted by Asian researchers but in non-Asian settings.

From the literature search, a total of 2295 articles were retrieved. These articles were screened by title and abstract by the two researchers of this study (WTT, CYO). Articles assessed were included or excluded according to the inclusion and exclusion criteria. Once the researchers finished their screening, they cross-checked each other's assessment for correctness. Discrepancies were resolved through discussions. From the 2295 articles, 2013 articles were removed as they did not fit the study inclusion criteria. The final number of articles included in this study were 282, and the corresponding authors' email addresses were extracted. Fig 1 illustrates the process of participant identification and the data collection process of this study.

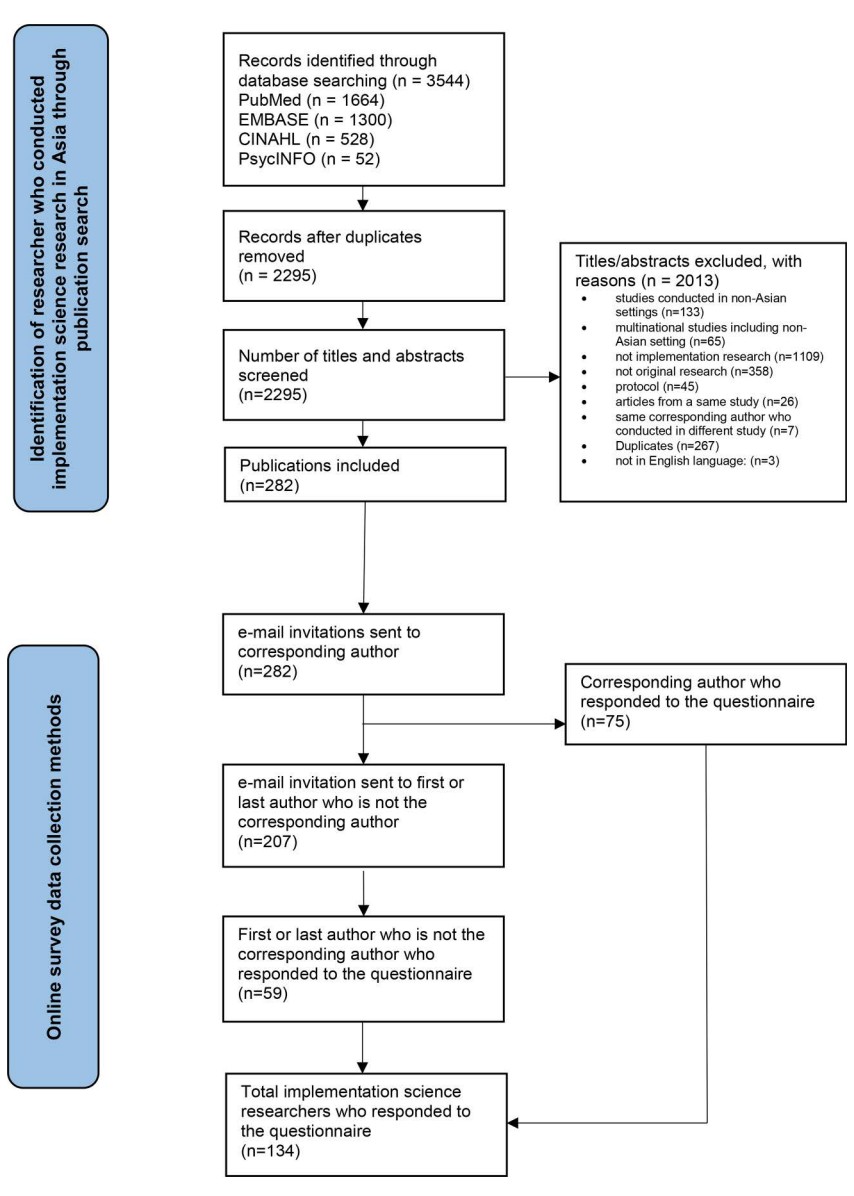

**Fig 1.  Study data collection process.**

## Instrument

The questionnaire in this study was adapted from a study conducted in the US on barriers towards engagement in IS research [22,23], literature review on barriers to conducting implementation science research [18,24–27], and expert input from four healthcare researchers who have conducted implementation science research in Asia. The questionnaire was sent to them with a rating scale on the relevance for each of the questionnaire item. They were also asked to provide comments in terms of clarity, and the significance and completeness of response options. Minor changes were made to the questionnaire such as revision of sentences that were unclear, and language and grammatical errors. The questionnaire was in English language and designed to be completed online using the RedCap electronic data capture tools hosted at the Universiti Malaya [28,29].

Subsequently, the questionnaire was pilot tested with 19 healthcare researchers who have experience with implementation research before finalising for actual study. No changes were made to the questionnaire. The finalised version of the questionnaire comprised of items that examine socio-demographic characteristics, implementation science research practices, and yes or no items which examines four barriers to conducting implementation science research that is, the lack of awareness, funding, time, and organisational support. An open-ended question was provided at the end of the questionnaire to gather additional information on barriers to IS research engagement that was not covered in the quantitative items (S1 File).

## Data collection process

The data collection for this study was conducted from April to June 2024. Participants were invited to participate in this study through email, where the RedCap survey link was included. Given that the participants were researchers who had conducted IS research in Asian settings as evidenced in the papers retrieved, their email address as a corresponding author was used. For participants who were not corresponding author, their email address was sought through Google search based on their affiliations given in the papers. The participant information sheet and the informed consent was embedded as part of the electronic questionnaire. Data confidentiality was ensured by informing the participants through the participant information sheet, that their data will be kept in the researcher's institution RedCap server, which provides a secure data storage environment. The participant ticked and dated the latest approved version of the informed consent form to proceed with answering the questionnaire.

Following the initial invitation email, two reminders were sent out: the first one a week later and the second one two weeks after the original invitation. If there were no responses after the second reminder, invitation was sent to the first or last authors, who were not corresponding authors of the study. Two reminders were also given. The online data collection process is illustrated in Fig 1.

## Data analysis

A descriptive analysis was conducted to summarize the results, utilizing frequencies and percentages with SPSS 28.0 (SPSS Inc, Chicago, IL) software. Responses to open-ended questions were analysed through qualitative content analysis, where data was condensed into categories based on the emerging findings. Initially, codes were developed to specific data sections that represented their meaning. Then, similar codes were grouped together to form categories. The broader categories represent the additional barriers to conducting IS research. The qualitative analysis was conducted by the researcher (WTT), and the findings were cross-checked with the other two researchers (CYO; YEL) to ensure accuracy. Any discrepancies were resolved through discussions.

## Ethics considerations

The study protocol was approved by the Universiti Malaya Research Ethics Committee (UMREC): UM.TNC2/UMREC_2989. Informed consent was obtained from all individual participants included in the study.

## Results

### Response rate

From the 134 authors who responded, only 113 questionnaires were included for analysis as 21 questionnaires were removed due to: incomplete response (n = 19), disagree to participate (n = 1), and duplicate entry (n = 1). The response rate for this study was 40.1% (113/282).

### Participants' characteristics

Many of the participants were between 35–44 years old (39.8%, n = 45). More than half have PhD level education attainment (68.1%, n = 77), and 46.0% (n = 52) were academic lecturers. There was also a considerable proportion of participants who were researchers (e.g., research officer, health service researcher, post-doctoral researcher) (24.8%, n = 28). Many of the participants' current primary institution were in Asia (69.0%, n = 72) while 31.0% (n = 41) were from the non-Asian countries such as US (14.2%, n = 16), UK (6.2%, n = 7), Australia (5.3%, n = 6), Canada (3.5%, n = 4), Sweden (2.7%, n = 3), Switzerland (1.8%, n = 2), France (0.9%, n = 1), and Germany (0.9%, n = 1). Slightly more than half of the participants have 10 years or less experience in their current position (55.7%, n = 63). Many of them have performed health-related research (including academic work) for 10 years or less (38.9%, n = 40), 11–20 years (33.6%, n = 48), and 21–30 (23.0%, n = 26) years.

### IS research practice

Majority of the participants reported that they have performed or collaborated on a study examining the translation of an intervention into routine settings in the past 5 years (85.8%, n = 97). Among those who have applied for funding for implementation research (n = 80), 78.8% (n = 63) were successful.

Among the countries within Asia, the most common country where IS research was conducted was in India (19.5%, n = 22), followed by Bangladesh (12.4%, n = 14), Nepal (10.6%, n = 12), and Indonesia (8.8%, n = 10). However, the most common country where the participants reported to have obtained funding for IS research was the US (15.9%, n = 18), followed by UK (9.7%, n = 8), India (7.1%, n = 8), Australia (5.3%, n = 6), China (5.3%, n = 6), Bangladesh (5.3%, n = 6), and Singapore (5.3%, n = 6) (Fig 2). Other funders (4.4%, n = 5) that were reported were Sexual Violence Research Initiative (SVRI), the Joint United Nations Programme on HIV/AIDS (UNAIDS), ViiV Healthcare, the International Association of Providers of AIDS Care (IAPAC), Gilead, the United States Agency for International Development (USAID), and the World Health Organization (WHO).

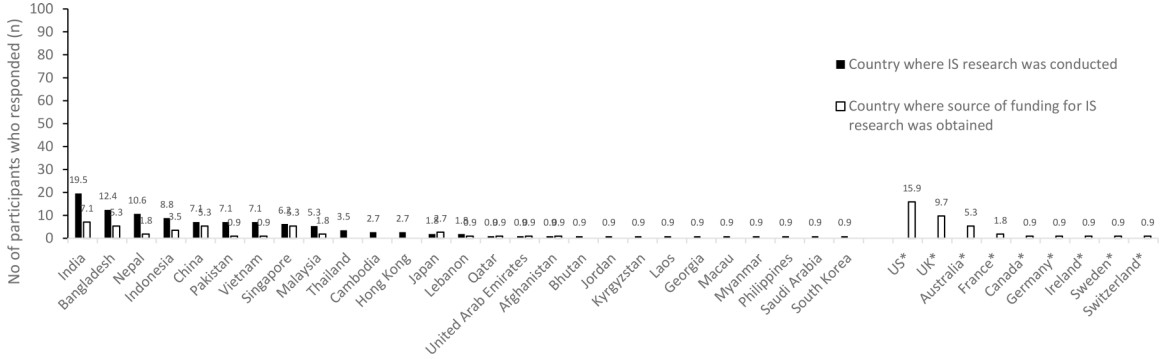

**Fig 2. The number of participants who reported the country in which they conducted IS research, and countries where they received funding from.**

## Barriers to conducting IS research

The most common barrier to conducting IS research was the lack of funding (86.7%, n = 98) followed by the lack of awareness (59.5%, n = 67), the lack of time (56.6%, n = 64), and the lack of organisational support (48.7%, n = 55), and Others (3.5%, n = 4) (Table 1). Findings from Others were incorporated together with comments from the open-ended question.

**Table 1. Barriers to conducting IS research and socio-demographic characteristics and IS practice.**

| | Total n (%) | Lack of awareness<br>Yes n (%) | Lack of funding<br>Yes n (%) | Lack of time<br>Yes n (%) | Lack of organi-sational support<br>Yes n (%) |
|---|---|---|---|---|---|
| **Age** | | | | | |
| 25 to 34 years | 11 (9.7) | 7 (10.4%) | 9 (9.2%) | 9 (14.1%) | 5 (9.1%) |
| 35 to 44 years | 45 (39.8) | 24 (35.8%) | 38 (38.8%) | 26 (40.6%) | 27 (49.1%) |
| 45 to 54 years | 30 (26.5) | 20 (29.9%) | 26 (26.5%) | 15 (23.4%) | 14 (25.5%) |
| 54 to 64 years | 27 (23.9) | 16 (23.9%) | 25 (25.5%) | 14 (21.9%) | 9 (16.4%) |
| | | | | | |
| **Highest educational level** | | | | | |
| PhD | 77 (68.1) | 47 (70.1%) | 68 (69.4%) | 44 (68.8%) | 40 (72.7%) |
| Clinical/professional masters | 15 (13.3) | 7 (10.4%) | 13 (13.3%) | 8 (12.5%) | 7 (12.7%) |
| Academic masters (Masters by research or coursework) | 15 (13.3) | 10 (14.9%) | 13 (13.3%) | 10 (15.6%) | 8 (14.5%) |
| Bachelor's degree | 6 (5.3) | 3 (4.5%) | 4 (4.1%) | 2 (3.1%) | 0 (0.0%) |
| | | | | | |
| **Current position** | | | | | |
| Academic lecturer | 52 (46.0) | 27 (40.3%) | 44 (44.9%) | 27 (42.2%) | 23 (41.8%) |
| Researcher (e.g., research officer, health service researcher, post-doctoral researcher) | 28 (24.8) | 18 (26.9%) | 24 (24.5%) | 20 (31.3%) | 18 (32.7%) |
| Policymaker/policy officer/senior management | 15 (13.3) | 8 (11.9%) | 15 (15.3%) | 8 (12.5%) | 7 (12.7%) |
| Healthcare practitioner/clinician | 10 (8.8) | 8 (11.9%) | 10 (10.2%) | 7 (10.9%) | 6 (10.9%) |
| Others (Postgraduate student (e.g.,: master, PhD) | 8 (7.0) | 6 (9.0%) | 5 (5.1%) | 2 (3.1%) | 1 (1.8%) |
| Duration of work in the current role? | | | | | |
| < 1–4 years | 32 (28.3) | 18 (26.9%) | 27 (27.6%) | 21 (32.8%) | 17 (30.9%) |
| 5 to 10 years | 31 (27.4) | 19 (28.4%) | 26 (26.5%) | 14 (21.9%) | 16 (29.1%) |
| 11 to 20 years | 29 (25.7) | 18 (26.9%) | 26 (26.5%) | 18 (28.1%) | 13 (23.6%) |
| 21 to 30 years | 17 (15.0) | 10 (14.9%) | 15 (15.3%) | 8 (12.5%) | 7 (12.7%) |
| 31 or more years | 4 (3.5) | 2 (3.0%) | 4 (4.1%) | 3 (4.7%) | 2 (3.6%) |
| Duration of having performed health-related research (including academic work) | | | | | |
| <1–4 | 4 (3.50) | 3 (4.5%) | 4 (4.1%) | 3 (4.7%) | 1 (1.8%) |
| 5–10 | 40 (35.4) | 21 (31.3%) | 29 (29.6%) | 24 (37.5%) | 17 (30.9%) |
| 11–20 | 48 (33.6) | 25 (37.3%) | 37 (37.8%) | 21 (32.8%) | 25 (45.5%) |
| 21–30 | 26 (23.0) | 14 (20.9%) | 23 (23.5%) | 11 (17.2%) | 9 (16.4%) |
| 31 or more | 5 (4.4) | 4 (6.0%) | 5 (5.1%) | 5 (7.8%) | 3 (5.5%) |
| | | | | | |
| Have performed or collaborated on a study examining the translation of an intervention into routine settings in the last five years | | | | | |
| No | 16 (14.2) | 10 (14.9%) | 14 (14.3%) | 8 (12.5%) | 8 (14.5%) |
| Yes | 97 (85.8) | 57 (85.1%) | 84 (85.7%) | 56 (87.5%) | 47 (85.5%) |

## Analysis of the open-ended responses

In addition to the commonly recognized barriers to conducting IS research—such as limited funding, awareness, time, and organizational support—open-ended responses revealed six additional challenges. These include insufficient commitment and engagement from stakeholders beyond researchers, a perceived lack of value attributed to IS research, concerns about the field's scientific rigour, challenges arising from existing knowledge, systems, and practices, inadequate capacity or training opportunities, and the absence of clear indicators for academic progression within the field (Table 2).

## Discussions

To our knowledge, this is the first study that assessed the barriers to conducting IS research in Asia. The study findings indicate that IS research is gaining momentum in the Asian region as many of the participants reported to have perform or collaborated on a study examining the translation of an intervention into routine settings in the past five years. Nevertheless, there was a high proportion who reported barriers to performing IS research. The prevalence of barriers reported in this study are higher compared to other studies conducted in the Western countries where the most prevalent barrier reported was the lack of funding (Asia: 86.7%; USA: 82.7%; multicountry study (Australia, Netherlands, Canada, USA, and the UK): 17.3%) followed by the lack of awareness (Asia: 59.5%; USA: 50.9%; multicountry study (Australia, Netherlands, Canada, USA, and the UK): 0.8%), and the lack of organisational support (Asia: 48.7%; USA: 28.9%; multicountry study (Australia, Netherlands, Canada, USA, and the UK): 9%) [22,24]. These barriers were also echoed in another study conducted in Western Europe countries namely Germany, Austria and Switzerland [25]. This study also found that many of the IS research was conducted among countries in South Asia specifically India, Bangladesh, and Nepal, while many reported to received funding from non-Asian countries such as from the UK and US.

Funding availability is a common challenge to conducting research in Asia. In majority of the Asian countries, the ratio of research and development expenditure to GDP is less than 1%, while more developed countries such as South Korea and China spent only about 4.5% and 2.1% respectively [22,30]. In this study, it was found that many participants received their

**Table 2. Additional barriers (n = 52).**

| Category | Selected quotes |
|---|---|
| Lack of capacity or training support (17.3%, n = 9) | • Weak capacities in the developing country governments to actively participate in the design, implementation, and dissemination of implementation science research, and to apply the learning for policy and programme improvements.<br>• Lack of a critical mass and not having IS topic as core public health education. |
| Lack of commitment and involvement from stakeholders other than researchers such as senior leaders, clinicians, or the government (15.4,%, n = 8) | • Resistance and poor understanding from stakeholders other than researchers regarding the multi-faceted efforts required for successful implementation, e.g., policy makers, management within health institutions.<br>• Difficulty in getting access to healthcare organisations to do the research.<br>• Lack of collaboration and support from the government. The slow and difficult process to work with government stakeholders. |
| Perceived low value of IS research (11.5%, n = 6) | • IS research is often not viewed as 'cutting edge' or transformational research among colleagues, academic institutions, and funding organisations.<br>• Studies focused on neurosciences, genetics, or artificial intelligence sound fancy while implementation science is not. |
| Existing knowledge, system and practices challenge (5.8%, n = 3) | • Getting support in real setting health care system is very difficult because it already has system and interventions<br>• As an applied field that draws on multiple disciplines and methodologies, implementation research grant applications often get stuck with purist and pedantic discipline-oriented peer reviewers. |
| IS lack of scientific rigour (3.9%, n = 2) | • IS often heavily relies on a qualitative approach, which many consider fluffy science. It lacks quantitative methods. |
| Lack of academic progression indicators (1.9%, n = 1) | • Translating research into practice often results in outputs that do not add to academic's career progression like a publication in a peer-reviewed journal would. It takes a lot more effort and time to do an implementation science activity versus doing a secondary data analysis that would lead to a peer-reviewed publication. |

funding to conduct IS research from the US and the UK. This is not surprising given that these countries have a longer IS history, and funding scheme to support dissemination and implementation research [31]. Some have also reported to receive funding from global organisation such as WHO, and other discipline-specific organisations such as UNAIDS. While these fundings support international research, this also meant that IS researchers in Asia are competing with researchers from other parts of the world on the pool of funding available to conduct IS research. Currently, dedicated funding for IS research in Asia is scarce. Much of the research funding in Asia prioritise basic science or research that produce commercialised innovations [32], as these types of research are deemed more prestigious and valuable [25]. To increase the availability of funding to conduct IS research in Asia, this will require priority shifting at the governmental level to set implementation research priorities. This can be observed in Singapore, and Hong Kong where IS is a priority area in their national research award and grant [33,34]. Apart from setting up dedicated fundings for IS, providing flexibility in the use of other discipline-specific research funding to support IS research, for example, as a secondary research question should be considered by funders [25].

The availability of funding will not increase IS research unless researchers have knowledge and skills to conduct IS research. Currently, IS training in the Asian region are scarce and sporadic, and this has impact on the ability of healthcare researchers in the region to effectively conduct implementation research [35]. Even when trainings are available locally, they are often conducted on ad hoc basis in the form of talks, or a few-days workshop, by inviting IS expert from the Western countries. Indeed, international network is a good first step in terms of IS training provision, but such can only be afforded by higher income countries in the region for example in Singapore, Hong Kong, and Malaysia. While there are online trainings on IS that can be accessible for free [36,37], these resources rely on self-study which may not be adequate to provide the necessary skills to design and perform high quality IS research [25]. There is a need to increase IS capacity building so that more training can be conducted locally. Another way to increase visibility and training especially among early career researchers is the incorporation of IS into the academic curricula [25]. Several master programme for IS has been developed in the region [38,39]. Conference offers another avenue for IS training, however, there is a lack of IS conference in Asia. Regional conferences that focus on implementation-related topic such as health systems and services (often under public health) do not include specific session on implementation science [40]. Cost is an issue for researchers from developing Asian countries to travel to the western counterparts [25] to attend IS conferences or to receive more formal IS trainings such as in the US, Canada or the UK [41]. Having an IS conference or event that specifically caters to the Asian community can increase IS visibility and understanding in the region, enable context specific topics to be discussed, formulation of agenda for establishing IS in the region [25].

One of the key approaches for effective and sustainable IS research is the involvement of stakeholders at the beginning and throughout the entire research [32]. However, this study revealed the lack of support and commitment from stakeholders (i.e., government stakeholders, senior leaders, or clinicians) as another barrier to conducting IS research. This barrier has also been expressed to be the biggest challenge for IS research as compared to other types of research as IS research requires willing practice partners from the implementation setting for the implementation to happen [42]. This barrier stemmed from the lack of awareness, perceived low value of the activity, and even hierarchical culture [43]. Building constructive long-term collaborations requires strategic approach, and this may not be something that many IS researchers are equipped with [25,26]. Strategies to increase the support and commitment from stakeholders in IS research that can be considered are: to have frequent communications with them to highlight the importance of IS research, making them feel exclusive by inviting them as expert panel rather as a participant, provide training to enhance their skills for effective participation, or publishing in peer-reviewed journals to show case credibility [43,44].

## Strengths and limitations

One notable strength of this study is that it is the first to examine the challenges associated with conducting implementation science research in Asian contexts. Secondly, this study assessed barriers to conducting IS research in Asia among participants who have conducted IS research in the Asian settings. The findings presented reflect real-life and not perceived

barriers. The limitations of this study include: the sampling frame is limited to researchers who has published their IS research in journals, hence, the findings presented may reflect those who primarily worked in the academic settings and may not be generalisable to IS researchers from non-academic or clinical settings; whose work are usually not visible to scientific community. As with any online survey, the response rate for this study is low. Although Internet access could pose a barrier for some participants, it is unlikely to be an issue for the target participants of this study, as the nature of their work typically requires them to have Internet access. Furthermore, since this study concentrates on the Asian region, it is presumed that the majority of the target participants will be from this area, where internet penetration exceeded 60% in 2024 [45]. The small sample size in this study rendered insufficient power to draw significant conclusions from the findings. Nevertheless, the descriptive findings provide valuable insights into barriers to conducting IS in the Asian settings.

## Conclusion

This study sheds light on the multifaceted barriers hindering the conduct of IS research in Asia, revealing a pressing need for systemic change to enable its advancement. The findings highlight that while IS research is gradually gaining traction in the region, it remains significantly constrained by limited funding, low awareness, insufficient training opportunities, and inadequate stakeholder engagement. Additionally, challenges such as the perceived low value of IS, lack of academic incentives, and systemic rigidity further impede progress. These barriers suggest that to advance IS in Asia, concerted efforts are required to establish dedicated funding streams, embed IS education within academic curricula, develop regional training programmes, and foster strategic, long-term collaborations with key stakeholders. Addressing these issues will not only build research capacity but also enable the generation of contextually relevant evidence to improve health outcomes and bridge the research-to-practice gap in diverse Asian healthcare settings.

## Supporting information

**S1 File. Search process and strategy, and study questionnaire.**
(PDF)

## Acknowledgments

The authors of this study would like to thank Dr Elizabeth R Stevens, Dr Donna Shelley, and Dr Bernadette Boden-Albala for their permission to adapt their questionnaire for this study. We are also thankful to Universiti Malaysia Sarawak for their financial support towards the Article Processing Charges (APC), which made the publication of this research possible.

## Author contributions

**Conceptualization:** Wen Ting Tong.

**Data curation:** Wen Ting Tong, Chor Yau Ooi, Yee En Lim.

**Formal analysis:** Wen Ting Tong.

**Funding acquisition:** Wen Ting Tong.

**Methodology:** Wen Ting Tong.

**Project administration:** Wen Ting Tong, Yee En Lim.

**Software:** Wen Ting Tong.

**Validation:** Wen Ting Tong.

**Writing – original draft:** Wen Ting Tong.

**Writing – review & editing:** Wen Ting Tong, Chor Yau Ooi, Yee En Lim.

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
