## [Decision Letter · Decision Letter 0]

PONE-D-25-11341Barriers to conducting implementation science research in Asia: An online surveyPLOS ONE

Dear Dr. Tong,

Thank you for submitting your manuscript to PLOS ONE. After careful consideration, we feel that it has merit but does not fully meet PLOS ONE’s publication criteria as it currently stands. Therefore, we invite you to submit a revised version of the manuscript that addresses the points raised during the review process.

Reviewer comments are attached and I invite you to consider these in your response. Both suggest providing more background and context for your study and strengthening the analysis and hence the impact on your conclusions. This will improve the clarity and potential impact of your work.

We look forward to receiving your revised manuscript.

Kind regards,

Jenny Wilkinson, PhD

Academic Editor

PLOS ONE

Journal Requirements:

“This study was funded by the Universiti Malaya Research Excellence Grant (UMREG2023_016)”

Reviewers' comments:

Reviewer's Responses to Questions

**Comments to the Author**

1. Is the manuscript technically sound, and do the data support the conclusions?

Reviewer #1: Yes

Reviewer #2: Partly

2. Has the statistical analysis been performed appropriately and rigorously? 

Reviewer #1: No

Reviewer #2: N/A

3. Have the authors made all data underlying the findings in their manuscript fully available?

Reviewer #1: Yes

Reviewer #2: No

4. Is the manuscript presented in an intelligible fashion and written in standard English?

Reviewer #1: Yes

Reviewer #2: Yes

5. Review Comments to the Author

Reviewer #1: Reviewer Comments

Introduction

Gaps in Literature and Justification

Although the intro section discusses barriers to IS research in Asia, explicit justification for why this particular study is necessary is visibly missing. The author should justify clearly why this study.

Scope and Specificity

Th authors try to share literature on IS research in high-income countries but do not compare same in other LMICs beyond Asia. At least, a broader contextualization by briefly mentioning IS research in another continent say in Africa or Latin America, would strengthen the argument.

Methods

Lack of Justification for Study Design

The authors clearly state that a cross-sectional online survey was used but do not justify why this design was chosen over the others. Also, the authors should briefly explain why an online survey is the most appropriate method for this study and a continent in the global south where online-related issues can sometimes be a challenge.

Participant Response Rate and Bias Consideration

The authors should briefly include a statement about the response rate and whether any follow-up measures were taken, like whether reminders were sent to encourage participation. This will help improve the bias considerations in the paper. The authors should clearly indicate how the questionnaires or surveys were administered. Was it through emails, a web-based platform as well as a narration on how security protection was ensured for electronic data? For example, if emails were used, did authors use their institution email addresses or personal email addresses etc?

The paper indicates that a 40.1% response rate was recorded. With this % it is easy for readers to perceive that the % of the non-respondents could affect the findings. So, the authors should briefly indicate where the non-respondents could have affected the findings. Results and

Discussion

I think the authors should advance the Statistical Analysis: The bivariate analysis is relatively basic. More advanced statistical techniques could have been used to explore relationships between key variables. Also, there is the absence of the use of numbers in the discussion. Authors should use a few numbers, or percentages to enhance the comparison hence strengthening the low patronage of IS in Asia. The numbers must speak to the barriers and the context. Though the quotes are good as stated in Table 2, the strength of the paper will improve if the authors could add the % of participants who mentioned each barrier.

The Conclusion is very short and lacks a statement on IS and its importance in research or practice

Reviewer #2: Thank you for the opportunity to review the manuscript titled “Barriers to Conducting Implementation Science Research in Asia: An Online Survey.“ Overall, the paper is well-written and easy to follow. I have provided some comments that could be addressed to enhance the transparency and implications of the study findings.

1. Please include a more substantial background stating that some work has been done on adapting the frameworks and ERIC strategies to the low-middle income context. For example:

• Means, A.R., Kemp, C.G., Gwayi-Chore, MC. et al. Evaluating and optimizing the consolidated framework for implementation research (CFIR) for use in low- and middle-income countries: a systematic review. Implementation Sci 15, 17 (2020). https://doi.org/10.1186/s13012-020-0977-0

• Lovero, K.L., Kemp, C.G., Wagenaar, B.H. et al. Application of the Expert Recommendations for Implementing Change (ERIC) compilation of strategies to health intervention implementation in low- and middle-income countries: a systematic review.Implementation Sci 18, 56 (2023). https://doi.org/10.1186/s13012-023-01310-2

• Kabir Sheikh, James Hargreaves, Mishal Khan, Sandra Mounier-Jack, Implementation research in LMICs—evolution through innovation, Health Policy and Planning, Volume 35, Issue Supplement_2, November 2020, Pages ii1–ii3, https://doi.org/10.1093/heapol/czaa118

2. On page 2, at the end of the introduction, the authors say, “The findings of this study will inform the development of strategies to promote IS research and ultimately increase the use of evidence-based interventions in the region.” The second part of this sentence is not a direct outcome of this study, so it sounds far-fetched. I recommend stating the direct implications of the findings, for example, demonstrating the challenges, raising awareness of the barriers, and ultimately identifying potential strategies to promote IS research further.

3. On page 3, the identification of the sample is very clear.

4. On page 4, the authors state, “The questionnaire in this study was adapted from a study conducted in the US on barriers towards engagement in IS research [18, 19], literature review, and input from healthcare researchers who have conducted implementation science research in Asia.” It would be helpful to describe what kind of literature review was conducted or to cite the manuscripts included in the review, how many researchers provided input, how that input was gathered, and what changes were made to the instrument based on the feedback. It is unclear if the input is the same as that from the pilot test involving 19 researchers.

5. On page 4, in the data analysis subsection, please provide details on how the qualitative analysis was conducted. For instance, how many comments were there? Was coding performed to identify categories? How many researchers analyzed the qualitative data? How were any discrepancies resolved? How many new types of barriers were identified through qualitative analysis?

6. Table 1: Given the small sample size across subgroups, I recommend presenting the results at a descriptive level and refraining from reporting associations, as these can lead to incorrect conclusions. For instance, a statistically significant association between the number of years of conducting health services research and barriers is notable, but it does not imply that other statistically non-significant associations are irrelevant. The study lacks sufficient power to draw such conclusions.

7. For transparency and replicability of the study, please include the instrument to the supplemental file.

6. PLOS authors have the option to publish the peer review history of their article (what does this mean? ). If published, this will include your full peer review and any attached files.

**Do you want your identity to be public for this peer review?** For information about this choice, including consent withdrawal, please see our Privacy Policy .

Reviewer #1: No

Reviewer #2: **Yes: ** Monisa Aijaz

---

## [Author Response · Author response to Decision Letter 1]

1 May 2025

Please refer to the Response To Reviewers file for the manuscript revision.

---

## [Editor Report · Decision Letter 1]

Barriers to conducting implementation science research in Asia: An online survey

PONE-D-25-11341R1

Dear Dr. Tong,

We’re pleased to inform you that your manuscript has been judged scientifically suitable for publication and will be formally accepted for publication once it meets all outstanding technical requirements.

Kind regards,

Jenny Wilkinson, PhD

Academic Editor

PLOS ONE
---

## [Editor Report · Acceptance letter]

PONE-D-25-11341R1

PLOS ONE

Dear Dr. Tong,

I'm pleased to inform you that your manuscript has been deemed suitable for publication in PLOS ONE. Congratulations! Your manuscript is now being handed over to our production team.

Kind regards,

on behalf of

Dr Jenny Wilkinson

Academic Editor

PLOS ONE